# Combining Recombinase-Mediated Cassette Exchange Strategy with Quantitative Proteomic and Phosphoproteomic Analyses to Inspect Intracellular Functions of the Tumor Suppressor Galectin-4 in Colorectal Cancer Cells

**DOI:** 10.3390/ijms23126414

**Published:** 2022-06-08

**Authors:** Malwina Michalak, Viola Golde, Dominik Helm, Herbert Kaltner, Johannes Gebert, Jürgen Kopitz

**Affiliations:** 1Department of Applied Tumor Biology, Institute of Pathology, Ruprecht-Karls-University Heidelberg, Im Neuenheimer Feld 224, 69120 Heidelberg, Germany; malwina.michalak@gmail.com (M.M.); viola.golde@web.de (V.G.); juergen.kopitz@med.uni-heidelberg.de (J.K.); 2Clinical Cooperation Unit Applied Tumor Biology, German Cancer Research Center, Im Neuenheimer Feld 280, 69120 Heidelberg, Germany; 3Proteomics Core Facility, German Cancer Research Center, Im Neuenheimer Feld 280, 69120 Heidelberg, Germany; d.helm@dkfz-heidelberg.de; 4Veterinary Faculty, Institute of Physiological Chemistry, Ludwig-Maximilians-University, 80539 München, Germany; kaltner@tiph.vetmed.uni-muenchen.de

**Keywords:** Galectin 4, proteomics, phosphoproteomics, colorectal cancer, Tet-On system

## Abstract

Galectin-4 (Gal4) has been suggested to function as a tumor suppressor in colorectal cancer (CRC). In order to systematically explore its function in CRC, we established a CRC cell line where Gal4 expression can be regulated via the doxycycline (dox)-inducible expression of a single copy wildtype *LGALS4* transgene generated by recombinase-mediated cassette exchange (RMCE). Using this model and applying in-depth proteomic and phosphoproteomic analyses, we systematically screened for intracellular changes induced by Gal4 expression. Overall, 3083 cellular proteins and 2071 phosphosites were identified and quantified, of which 1603 could be matched and normalized to their protein expression levels. A bioinformatic analysis revealed that most of the regulated proteins and phosphosites can be localized in the nucleus and are categorized as nucleic acid-binding proteins. The top candidates whose expression was modulated by Gal4 are PURB, MAPKAPK3, BTF3 and BCAR1, while the prime candidates with altered phosphorylation included ZBTB7A, FOXK1, PURB and CK2beta. In order to validate the (phospho)proteomic data, we confirmed these candidates by a radiometric metabolic-labelling and immunoprecipitation strategy. All candidates exert functions in the transcriptional or translational control, indicating that Gal4 might be involved in these processes by affecting the expression or activity of these proteins.

## 1. Introduction

Galectins are members of a family of 15 carbohydrate-binding proteins that contain one or two carbohydrate-binding domains specific for galactose-containing glycans. Galectins are widely expressed in various cell types and mediate their functions both intra- and extracellularly. They are involved in the regulation of cell growth, differentiation, signal transduction, apoptosis, pre-mRNA splicing, cell–cell and cell–matrix interactions, cellular polarity, motility and innate/adaptive immunity [1,2]. Considering their important physiological functions, it is not surprising that an abnormal expression of galectins has been implicated in several disease states. In particular, galectins have been shown to play a fundamental role in cancer development, progression and metastasis [2,3,4]. Galectin-4 (Gal4; gene name *LGALS4*) is primarily expressed in epithelial cells of the gastrointestinal tract. It belongs to the tandem-repeat subfamily of galectins, which are composed of two identical carbohydrate recognition domains (CRDs) at the N-terminus and C-terminus joined by a short linker sequence (the “hinge” region). The CRDs distinctly differ in the binding preferences for their ligands, which encompass glycosphingolipids, oligosaccharides and glycoproteins bearing lactosyl structures. The physiological functions of Gal4 are not fully resolved, but their roles in the stabilization of lipid rafts, cell adhesion, apical trafficking in enterocytes and intestinal wound healing have already been established [5]. These examples suggest galectin’s broad intracellular activity, as well as its functionality at the cell surface. Likewise, the roles of Gal4 in tumor development and progression may also be ascribed to intra- and extracellular actions. For instance, the loss of cell surface binding of Gal4 has been shown to promote colorectal tumorigenesis by affecting the expression of the genes involved in growth control and apoptosis [6,7], while the addition of Gal4 to the culture medium of a set of colorectal cancer cell lines profoundly reduced their proliferation rates and induced differentiation. In these detailed proteomic and phosphoproteomic analyses, broad effects of exogenous Gal4 on the cell’s molecular phenotype, due to expression changes in a large number of proteins, are indicated. The bioinformatic interpretation of these results by protein ontology analyses supports the concept of Gal4`s action as a tumor suppressor [8]. Moreover, additional phosphoproteomic analyses revealed that exogenous Gal4 has a bearing on malignancy-associated intracellular protein phosphorylation [9].

Besides their presence in the extracellular milieu and on the outer surface of cells, galectins have been detected in relatively high concentrations intracellularly, e.g., in the cytosol or the nucleus [10]. While the binding of galectins to the cell surface is considered to be mediated by binding to membrane-anchored surface receptors via their CRDs, intracellular actions may involve both CRD-dependent and -independent mechanisms [11]. In colorectal, pancreatic and hepatocellular cancer, intracellular Gal4 appears to exert tumor-suppressing activity by the inhibition of protumorigenic signaling pathways [12,13,14]. However, how the loss of Gal4 expression in CRC supports tumorigenesis on the molecular level is still poorly understood. 

In order to systematically address this topic, we now established a CRC cell line where Gal4 expression can be regulated via the doxycycline (dox)-inducible expression of a single copy wildtype *LGALS4* transgene generated by recombinase-mediated cassette exchange (RMCE). Using this model system and pursuing a quantitative proteomics and phosphoproteomics approach, we sought to identify target proteins whose expression and phosphorylation is regulated specifically by Gal4.

## 2. Results

### 2.1. Generation of Colorectal Cancer Model Cell Line with Dox-Inducible Gal4 Expression

In order to investigate the role of the expression loss of Gal4 in CRC, we established a colorectal cancer cell line model system that enables the inducible reconstitution of Gal4 expression in an isogenic background. The model cell line is derived from Gal4-deficient cell line HCT116 by recombinase-mediated cassette exchange (RMCE). The HCT116 HygTK master cell line carries a Hyg-TK expression cassette flanked by two recombination sites (F3/F), which confers hygromycin resistance. Two independent clones of this master cell line have been established in our lab previously, differing in the integration site of the expression cassette, which is either localized on chromosome 1 (*C1orf159*) for clone #5 and on chromosome 5 (*ALDH1L1* gene, silenced by methylation in HCT116 cells) for clone #22 [15,16]. The luciferase-*Gal4* expression cassette was inserted directly into these defined loci by recombinase-mediated cassette exchange, resulting in HCT116-Gal4 #5 and #22 cells. In these model cell lines, Gal4 and luciferase expression are regulated by the bidirectional dox-inducible promoter (P_tet_bi) that is comprised of two minimal promoters, human cytomegalovirus (hCMV) and mouse mammary tumor virus (MMTV)-2. This promoter allows the initiation of gene transcription in the presence of doxycycline (dox) by allowing the binding of the reverse transcriptional transactivator to seven Tet operators within the promoter construct (Figure 1A, Tet-On system). The established cell line therefore enables the specific investigation of the biological relevance of the loss of Gal4 in CRC tumorigenesis. Two independent clones, showing strong dox-inducible luciferase activity, as well as Gal4 expression on the protein level, were chosen for detailed characterization and further study (Figure 1B–D).

### 2.2. Characterization of Doxycycline-Inducible HCT116-Gal4 Cell Lines

The induction of Gal4 expression in two independent monoclonal cultures of HCT116 #5 and #22 was examined in a dose- and time-dependent manner. Each clone showed a slightly different dose-dependency from Gal4 expression. Clone #22 showed a higher induction in luciferase activity than #5 on the protein level, while clone #5 exhibited a more efficient induction of Gal4 re-expression (Figure 2A,C). For clone #5, the induction of Gal4 expression was on a comparable level in the range of 0.25–1 µg/mL Dox. In all further experiments, a Dox concentration of 0.5 µg/mL was chosen for HCT116 Gal4 #5 cells to ensure a strong Gal4 expression. Clone #22 shows stable and strong Gal4 expression at about 0.75 µg/mL Dox and higher. Consequently, 0.75 µg/mL of Dox was used for all subsequent experiments with this clone. The time–course analysis of both model cell lines showed that Gal4 was detectable after 12 h and reached a peak within 72 h. In the absence of Dox, no Gal4 protein was detectable (Figure 2B,D). Proliferation assays performed on both model cell line clones, as well as the master cell line HCT116-HygTk and parental HCT116 cells, showed no significant influence of the chosen doxycycline concentrations on cell growth. Similarly, reconstituted Gal4 expression did not have any effect on the cell proliferation rate of these cells (Appendix A). In order to check for Gal4 secretion to the extracellular space, we examined the presence of Gal4 in conditioned media of both HCT116-Gal4 clones. In addition to the presence of Gal4 in lysates of dox-treated cells, Gal4 was also clearly detectable in and, thus, secreted onto their medium upon dox treatment (Figure 3), whereas it was completely absent in the medium of the untreated cells.

### 2.3. Gal4 Induces Proteomic and Phosphoproteomic Changes

As both clones of our model cell line HCT116 Gal4 showed similar growth and expression characteristics, only one clone (HCT115-Gal4 #5) was chosen for the subsequent detailed analyses of Gal4-induced proteomic and phosphoproteomic changes. For this purpose, the SILAC (stable isotope-labeling with amino acids in the cell culture) approach for mass spectrometry-based proteomics and phosphoproteomics was applied. These experiments were conducted in three biological replicates. HCT116 Gal4 #5 cells were labeled with “heavy medium” (containing L-[^13^C_6_, ^15^N_4_] arginine (R10); L-[^13^C_6_, ^15^N_2_] lysine) or “light medium” (containing L-[^12^C_6_,^14^N_4_] arginine (R0); L-[12C6,14N2] lysine). After exhaustive labeling for 8 days, the ‘heavy’ cells were treated with doxycycline for 72 h for induction of Gal4 expression, while ‘light’ cells served as the untreated control (Appendix A). A total of 3083 proteins were identified and quantified in at least two biological replicates (Appendix A). Twenty proteins were regulated (fold change > 1.5) in a Gal4-dependent manner (Table 1).

In parallel, 2071 phosphorylation sites were identified with a high probability (>0.75) and quantified in at least two biological replicates. To each phosphosite, corresponding information from the PhosphoSite Plus database was aligned (Appendix A). In order to avoid the false identification of phosphorylation changes due to alterations in whole protein expression, 1603 phosphosites were matched and normalized to their protein expression levels (Appendix A). Among them, 48 phosphosites, located on 21 phosphoproteins, were found to be regulated (>1.5-fold change) in at least two biological replicates (Table 2).

When the regulated proteins and phosphoproteins were mapped together and visualized by the STRING database in an interaction network extracted from the entire human genome (Figure 4A), the resulting network consisted of 540 nodes and 9808 edges. A Gene Ontology enrichment analysis showed strong enrichment in proteins involved in RNA binding (*p* = 9.23 × 10^−126^) and nucleic acid binding (*p* = 1.29 × 10^−111^). The enrichment analysis also revealed that most of the regulated proteins and phosphoproteins can be localized in the nucleoplasm (*p* = 1.24 × 10^−127^), nuclear lumen (*p* = 7.16 × 10^−127^) and nucleus (*p* = 5.26 × 10^−114^).

In order to check the validity of the data obtained by quantitative (phospho)proteomics, we selected four representative candidates, each from the proteomic and phosphoproteomic analysis for validation by an independent method. To get a sensitive and reliable quantitation of the candidates, we combined radiometric metabolic labeling with immunoprecipitation. A dual metabolic labeling strategy was applied to account for potential alterations in protein expression on the data obtained with a phosphoproteomic analysis. To this end, ^35^S-methionine was used to label the protein part, while ^32^P-orthophospate was applied to detect their modification by phosphorylation. The Gal4-dependent upregulation of PURB (transcriptional activator protein Pur-beta) and MAPKAPK3 (MAPK-Activated Protein Kinase 3) expression, as well as the downregulation of BTF3 (Basic Transcription Factor 3) and BCAR1(Breast Cancer Anti-Estrogen Resistance Protein 1), could be confirmed (Figure 5). 

Likewise, the altered phosphorylation of ZBTB7A (Zinc Finger and BTB Domain Containing 7A), FOXK1 (Forkhead Box K1), PURB and CK2beta (Casein Kinase II subunit beta) could be corroborated (Figure 6). Interestingly, PURB was upregulated at the protein expression level, accompanied by a decreased phosphorylation level.

## 3. Discussion

In previous studies, the transient transfection of *LGALS4* into pancreatic carcinoma [13], gastric cancer cells [17] oligodendrocytes [18] and CRC cell lines [7] was already used to investigate the intracellular functions of Gal4. In order to overcome the disadvantages of these hitherto applied transfection strategies, we, for the first time, applied a RCME-based strategy and inducible target gene expression to test Gal4 functions in a cancer cell line, which brings some benefits. In particular, only a single copy of the *LGALS4* transgene is integrated into the genome of the model cell line. The expression of the transgene can be regulated by dox, thereby allowing the induction of the physiological levels of Gal4 in an isogenic background. The careful characterization of two independent HCT116-Gal4 clones proofed the inducibility of Gal4 expression in a time- and dose-dependent manner. Thus, the engineered HCT116-Gal4 cell lines provide a versatile tool to analyze functional consequences and (patho)biological relevance of altered Gal4 expression in CRC.

The induction of intracellular Gal4 expression did not affect the growth rate of the cells. By contrast, Satelli et al. [7] described a reduced proliferation after an overexpression of Gal4 in HCT116 cells. This might be explained by the different gene doses of our model system versus the transient transfection strategy. Moderate amounts of Gal4 were detectable in the conditioned medium of HCT116Gal4 #5 and #22, indicating that the galectin is secreted by the model cell lines. Since the secretion of galectins is mediated independently from the classical secretory pathway by a not yet fully defined nonclassical pathway [19,20], our cell lines may become a versatile tool to address this in future studies. The binding of Gal4 to the extracellular surface of CRC cells has been shown to exert a strong inhibitory effect on proliferation [8]. However, the secretion rates in our model system obviously do not generate extracellular concentrations that are sufficient to affect cell growth.

In order to systematically screen for intracellular changes induced by Gal4 expression, we quantified the proteomic and phosphoproteomic changes in our model system. Broad coverage of the cellular proteome was achieved by quantitating more than 3000 cellular proteins. In parallel, over 2000 phosphorylation sites were identified and quantified. Combining proteomic and phosphoproteomic analyses brought up the advantage that alterations in protein expression could be taken into account in the protein phosphorylation analysis. A bioinformatic evaluation of the results suggests a role of Gal4 in nucleic acid binding. Accordingly, the transcriptional activator protein Purbeta [21] and MAP kinase-activated protein kinase 3 [22] (MAPKAPK3), which plays an essential role in modulating gene transcription in the nucleus, are among the top upregulated proteins. 

Other signaling proteins involved in transcriptional control include basic transcription factor 3 (BTF3) [23] and breast cancer anti-estrogen resistance 1 (BCAR1) protein [24], which are downregulated upon Gal4 expression. For validation of the proteomic results, we focused on these proteins, since there were hints of their potential function in cancer. For this purpose, we chose combining radiometric metabolic labeling with immunoprecipitation, since this method allows exact quantitative measurements. Moreover, applying this approach to both HCT116-Gal4 clones underscores the validity of the data. Thus, this independent approach corroborates the proteomic data.

MAPKAPK3 is part of the p38 mitogen-activated protein kinase (p38MAPK) pathway. It has a high structural identity with MAPKAP2 and shares the same substrate specificity, suggesting very similar functions in biological systems. MAPKAP2 is considered a master regulator of RNA-binding proteins. However, the specific functions of MAPKAPK3 are not clearly defined yet [25]. Tumorigenic and also tumor-suppressive functions of MAPKAPK3 have been described. In cancer cell models, its loss provided a growth advantage [26]. BTF3 has been described as an oncogene in various cancers. In CRC tissue, BTF3 is overexpressed and correlates with a poor prognosis [27]. A recent study indicates that the oncogenic action in colon cancer is linked to an induction of a DNA helicase (CHD1L) and inhibition of E3 ubiquitin ligase HERC2-mediated p53 degradation [28]. BCAR1 belongs to the family of CAS proteins, which are overexpressed in many solid tumors, where they promote invasion and metastasis [29,30].

The most pronounced effect observed upon reconstituted Gal4 expression was an approximately 10-fold upregulation of ZBTB7A phosphorylation at serine residue 549. The protein acts as a pleiotropic transcription factor that is functionally involved in various stages of cell proliferation and differentiation. In addition, ZBTB7A overexpression has also been linked to tumorigenesis and metastasis in various cancer types, including breast, prostate, lung, ovarian and colon cancer [31]. The S549 phosphorylation site has already been described [32], but its physiological significance is not yet known. The new finding of its prominent Gal4-dependent regulation points to the possibility that it might be involved in the control of ZBTB7A activity, which requires future functional studies. FOXK1 is a transcription factor that regulates the expression of many genes, thereby controlling various cellular functions, including cell cycle, cell growth, proliferation, apoptosis, autophagy, stress resistance, metabolism, DNA damage, drug resistance, angiogenesis and carcinogenesis [33]. FOXK1 is a substrate of glycogen synthase kinase 3 (GSK3), and the phosphorylation of FOXK1 regulates its translocation between the cytoplasm and nucleus. FOXK1 carries a variety of phosphosites that may be involved in its regulation (www.phosphonet.ca, accessed on 1 April 2022). The phosphorylation of three sites in FOXK1 (T436, S441 and S445) is upregulated by Gal4. In a study that quantitated all phosphosites of FOXK1 during mitotic cell proliferation, all three sites were already listed, but they were not changed during mitosis [34]. CK2beta is the regulatory subunit of the tetrameric CK2 protein complex that is involved in various cellular processes, with a focus on the control of cell growth, proliferation and survival. Its activity is increased in nearly all cancers, and upregulated CK2 is considered a key player in cancer biology [35]. CK2beta modulates the enzymatic activity and substrate specificity, as well as the assembly of the CK2 complexes [36]. In turn, CK2beta is regulated by phosphorylation/autophosphorylation [37,38]. However, the influence of phosphosite S228 is not yet defined. The expression of transcriptional activator protein PURB was upregulated by Gal4, while its phosphorylation at sites S6 and S8 was decreased. It belongs to the PUR protein family that are involved in the initiation of DNA replication, cell cycle regulation and in the regulation of transcription and RNA translation [39]. The expression or activity of PURB is altered in several cancers [40,41]. Its phosphorylation at S6 and S8 was described in a previous study [42]. The observation that an increased expression of PURbeta correlates with decreased phosphorylation might be considered as a hint that phosphosites S6 and S8 are involved in the Gal4-controlled stability of PURB.

Altogether, our cell line model allows the targeted analysis of intracellular Gal4 functions, as exemplified by our proteomic and phosphoproteomic analyses. Our study suggests a nuclear function of Gal4 in transcription control and defines promising candidates for future research. However, each candidate—in particular, those that have been confirmed by metabolic labeling/immunoprecipitation—has to be pursued in extensive follow-up studies. Nevertheless, the present study highlights the potential of our experimental strategy to set starting points and to direct future research on a specific protein with potentially important but still poorly defined functions, such as Gal4. It may be adapted to any other protein of interest.

## 4. Materials and Methods

### 4.1. Cancer Cell Lines and Culture Conditions

Colorectal cancer cell lines were grown under the standard conditions in DMEM/F12 medium (Thermo Fisher Scientific, Darmstadt, Germany) supplemented with 10% FCS in the presence of 100 U/mL penicillin and 100 µg/mL streptomycin (PAA Laboratories GmbH, Cölbe, Germany). HCT116 (ECACC#91091005), Colo 205 (ECACC#87061208) and HeLa (ECACC#93021013) cells were obtained from the European Collection of Authenticated Cell Cultures (ECACC; https://www.phe-culturecollections.org.uk (accessed on 7 January 1997). HCT116 HygTK cell line was generated previously in our laboratory [15]. All cell lines were regularly checked for mycoplasm contamination using the Mycoplasma Detection Kit (Minerva Biolabs, Berlin, Germany). Cell growth was determined by cell counting in a Neubauer hemocytometer.

### 4.2. Nucleic Acid Isolation, Analysis and RT-PCR

Plasmids were purified using Plasmid Mini, Midi or Maxi Kits (Qiagen, Hilden, Germany) following the manufacturer’s instructions. Human full-length *LGALS4* cDNA was PCR-amplified by Q5 High-Fidelity DNA Polymerase (New England BioLabs, Frankfurt am Main, Germany) from plasmid pGEMEX-GAL4 using primers with EcoR1 and Not1 restriction sites and the following cycle conditions: 98 °C for 30 s, followed by 40 cycles at 98 °C for 10 s, 65 °C for 30 s, 72 °C for 40 s and a final extension at 72 °C for 5 min. PCR amplicons were double-digested with EcoR1-HF (100,000 U/mL; New England BioLabs) and Not1-HF (20,000 U/mL; New England Biolabs). Plasmid pcDNA 3.1-*Gal4* was generated by ligation of the EcoR1/Not1 *Gal4* PCR fragment into EcoR1/Not1 digested and dephosphorylated plasmid pcDNA3.1 (Rapid DNA Dephos & Ligation Kit, Roche, Mannheim, Germany) and transformation into bacterial strain DH5a. Gal4-positive bacterial clones were identified by colony PCR using the following cycle conditions: activation of DNA polymerase (HOT MOLPol; projodis GmbH, Butzbach, Germany) at 95 °C for 30 s, followed by 35 cycles at 95 °C for 30 s, 60 °C for 30 s and 72 °C for 1 min, with a final extension at 72 °C for 6 min. Finally, the retroviral vector S2F-cLM2CG-FRT3-Gal4 was generated by subcloning the *Eco*R1/*Not*1 *Gal4* fragment from pcDNA3.1-*Gal4* plasmid into *Eco*R1/*Not*1 digested and dephosphorylated retroviral expression plasmid S2F-cLM2CG-FRT3 [43], thereby replacing the *Eco*R1/*Not*1 *mCherry* fragment. Verification of the *LGALS4* sequence in pcDNA 3.1-*Gal4* and S2F-cLM2CG-FRT3-*Gal4* was confirmed by DNA sequencing. All used primers are listed in Appendix A.

### 4.3. Generation and Characterization of HCT116-Gal4 Cell Line

HCT116-Gal4 cells were generated by using the master cell line HCT116-HygTK, which is hygromycin-resistant but sensitive to ganciclovir [15]. The 8.8 × 10^6^ cells were electroporated with 3 μg of S2F-cLM2CG-FRT3-Gal4 and 3 μg of pCAGGS-Flpo-IRES-Puro plasmid using the AmaxaCell Line Nucleofactor Kit V (Lonza, Basel, Switzerland), according to the manufacturer’s instructions. Upon RMCE, the integrated HygTK expression cassette was replaced by an expression cassette encoding for luciferase and Gal4. After 24 h, 1.5 μg/mL puromycin (Sigma-Aldrich, Taufkirchen, Germany) was added to the growth medium for 36 h, followed by a selection with 40 μM Ganciclovir (Roche, Mannheim, Germany) for at least 2 weeks. Doxycycline-inducible Gal4 expression in individual clones was confirmed by luciferase activity measurement and Western blot analysis. For reconstitution of Gal4 expression, cells were either grown in the presence or absence of 0.5 μg/mL doxycycline (Dox) (Sigma-Aldrich) for 24 h.

To investigate the dose- and time-dependent inducible expression of *LGALS4*, cells were incubated with different concentrations of Dox (0.01–1 µg/mL) for 24 h and further grown in the presence or absence of 0.5 μg/mL (clone #5) and 0.75 μg/mL (clone #22) Dox for 6 h and up to 72 h, respectively. Dox concentration and time of treatment were chosen for each clone separately in order to ensure a high Gal4 expression.

To analyze the secretion of Gal4, cells were grown in the presence or absence of 0.5/0.75 μg/mL Dox. After 48 h, the cells were washed twice with DMEM/F12 medium, and cells were cultured grown in the presence or absence of 0.5/0.75 μg/mL Dox for the next 24 h in DMEM/F12 medium supplemented with Insulin–Transferrin–Selenium–Ethanolamine (Thermo Fisher Scientific), according to the manufacturer’s instructions. The collected medium was further concentrated on a centrifugal filter, Amicon Ultra-15 (MWCO 10,000, GE Healthcare, Münster, Germany), to a final volume of 0.5 mL. A sample (16.25 µL) of concentrated cell culture medium was analyzed by Western blot. 

### 4.4. Luciferase Assay

Cells were grown on 96-well plates in the presence or absence of 0.5–0.75 μg/mL Dox for up to 72 h. Proliferation was assessed using the CellTiter 96^®^ Aqueous One Solution Cell Proliferation Assay (Promega), according to the manufacturer’s protocols.

### 4.5. Proliferation Assay

Cells were grown on 96-well plates in the presence or absence of 0.5–0.75 μg/mL Dox for up to 72 h. Proliferation was assessed using the CellTiter 96^®^ Aqueous One Solution Cell Proliferation Assay (Promega), according to the manufacturer’s protocols. 

### 4.6. Western Blot Analysis

Protein was extracted by lysing cell pellets in RIPA buffer containing 50 mM Tris-HCl (pH 7.5), 150 mM NaCl, 1% Triton X-100, 0.5% sodium deoxycholate and 0.1% SDS supplemented with fresh protease (cOmplete Mini; Roche), followed by sonication (Bandelin-Sonopuls, Bandelin Electronic GmbH & Co. KG, Berlin, Germany) and centrifugation (13,000× *g*, 30 min, 4 °C). The protein concentration was determined by the Bradford assay, and 50 µg of protein were subjected to a Western blot analysis, as described in [8], with anti-Gal4 (1:2000, rabbit-polyclonal, kindly provided by Prof. Gabius, LMU, Munich), followed by secondary anti-rabbit-HRP Conjugate (7074S, Cell Signaling Technology, Danvers, MA, USA). Visualization was performed with Western Lighting Chemiluminescence Reagent Plus (Perkin-Elmer LAS, Inc., Waltham, MA, USA) using a ChemiDoc™ MP System (Bio-Rad, Munich, Germany). β-actin detection, using a mouse monoclonal anti-actin antibody (3700S, 1:1000, 1 h at RT; Cell Signaling Technology, Danvers, MA, USA) and anti-mouse-HRP as the secondary antibody (NXA931, 1:5000; GE Healthcare, Buckinghamshire, UK), served as the loading control.

### 4.7. SILAC Labeling and Protein Extraction

SILAC labeling was performed as described previously [8,9]. Briefly, two cell populations of model cell line HCT116-Gal4 #5 were cultured separately in ’heavy’ medium (containing L-[^13^C_6_, ^15^N_4_] arginine (R10); L-[^13^C_6_, ^15^N_2_] lysine (K8)) or ‘light’ medium (containing L-[^12^C_6_,^14^N_4_] arginine (R0); L-[^12^C_6_,^14^N_2_] lysine (K0)) (Silantes, München, Germany) for 8 days. To avoid arginine-to-proline conversion, the medium was additionally supplemented with L-proline (Sigma-Aldrich) to a final concentration of 200 µg/mL [8]. ‘Heavy’ labeled cells were then treated for 72 h with doxycycline (500 ng/mL) to induce the Gal4 expression, whereas the control cell populations were exposed to dox-free medium (heavy and light labels still present in the medium). The experiment was performed in triplicate. For protein extraction, cells were suspended in RIPA buffer supplemented with fresh protease (cOmplete Mini; Roche, Mannheim, Germany) and phosphatase (PhosSTOP, Roche, Mannheim, Germany) inhibitors and treated with benzonase (125 U; Merck, Darmstadt, Germany) on an orbital shaker (at 300 rpm) on ice for 1 h. After centrifugation at 13,000× *g* for 30 min at 4 °C, the protein concentration of the extracts was measured by using 2D Quant Kit reagents (GE Healthcare, Buckinghamshire, UK), according to the manufacturer’s instructions. 

### 4.8. Tryptic Digestion

Protein lysates from both culture conditions (‘heavy’ and ‘light’) were mixed in a 1:1 ratio based on their protein concentration (in total 410 µg). Quantitative protein precipitation using a methanol–chloroform–water mixture [44] was performed in order to remove reagents, especially protease inhibitors, prior to tryptic digestion. Precipitated protein samples were redissolved in 150 µL digestion buffer (6 M urea and 100 mM NH_4_HCO_3_) by incubation at 25 °C for at least 1 h. Then, samples were first treated with 5 µL of 1 M dithiothreitol (DTT) in digestion buffer at 45 °C for 1 h to completely reduce the disulfide bonds, followed by the alkylation of thiol groups by adding 7 µL of 0.5 M iodoacetamide (IAA) in digestion buffer and 30 min of incubation in the dark at 25 °C. The remaining free IAA was blocked by adding 5 μL of 1M DTT-containing solution and incubation for 15 min at 37 °C. Pre-digestion was performed with 2.5 μg Lys-C (Promega GmbH, Heidelberg, Germany) in 100 mM NH_4_HCO_3_ overnight incubation at 37 °C with constant gentle shaking. Then, the samples were diluted with 700 µL of 100 mM NH_4_HCO_3_ and 0.125 mM CaCl_2_ to achieve the UREA concentration (~1 M) required for the perseverance of tryptic activity. Tryptic digestion was conducted overnight at 37 °C with 6.5 μg trypsin (Promega GmbH, Heidelberg, Germany). After digestion, 10 µg of the protein digested were subjected to a shotgun mass spectrometry analysis, while 400 µg protein digest underwent phosphopeptide enrichment for a phosphorylation analysis. 

### 4.9. Phosphopeptide Enrichment

In order to clean and concentrate the peptide mixtures after tryptic digestion, the StageTip procedure [45] was applied using C18 material and the reversed phase material (Oligo™ R3, Thermo Fisher Scientific, Darmstadt, Germany) packed into a pipette tip (volume up to 200 µL). Briefly, binding was performed with 2.5% formic acid, followed by washing with 2.5% formic acid and elution with 2 times 100 μL of 0.6% acetic acid in 80% acetonitrile. The samples were dried in a vacuum centrifuge.

Each sample was then resuspended in 300 µL binding buffer (80% acetonitrile (ACN), 5% trifluoroacetic acid (TFA) and 0.2% glycolic acid) by sonication and 30 min of incubation on a Thermoshaker (250 rpm, RT) and added to 100 µL of freshly prepared 5% PureCube Fe^3+^-NTA MagBeads (31505-Fe; Biozym Scientific GmbH, Hessisch-Oldenburg, Germany) in the binding buffer. Samples were incubated with the beads for 1.5 h on a rotator at RT. Subsequently, the beads were washed twice with 80% ACN/1% TFA, followed by a wash with 10% ACN/0.2% TFA. Phosphopeptides were eluted by 30 min of incubation with 80 µL of 1% ammonium hydroxide (250 rpm, RT). The samples were then purified by the StageTip procedure, as described above. Each solution was dried completely in a vacuum centrifuge and frozen. 

### 4.10. LC-MS/MS

Dried peptides were reconstituted prior to LC-MS/MS analysis in 2.5% Hexafluoro-2-propanol, 0.1% TFA in water or 50 mM citrate and 0.1% TFA, respectively. First, the peptides were loaded onto a trapping cartridge (Acclaim PepMap300 C18, 5µm, 300 Å wide pore, Thermo Fisher Scientific, Darmstadt, Germany) and desalted for 3 min using 0.05% TFA. Peptide separation was performed using a multistep gradient of buffer A (0.1% formic acid in water) and buffer B (0.1% formic acid in acetonitrile), with the main step ramping up the buffer B concentration from 5% to 30% (28% for phosphopeptides) over 132 min on a nanoEase MZ Peptide analytical column (300 Å, 1.7 µm, 75 µm x 200 mm, Waters) using an UltiMate 3000 UHPLC system (60 min of total analysis time). Eluting peptides were subsequently analyzed by an Orbitrap Exploris 480 mass spectrometer (Thermo Fisher Scientific, Darmstadt, Germany) running in data-dependent acquisition mode, where one full scan (380–1400 *m*/*z*, 300% AGC target, maxIT 45 ms) at 120-k resolution was followed by MS/MS scans for 2-s cycle times. Precursors were isolated with 1.4 *m*/*z* (1.2 *m*/*z*), peptides were fragmented using 26 NCE (28 NCE) and MS/MS scans were recorded at 15-k resolution (100% AGC target, maxIT 22 ms; 200% AGC, maxIT 54 ms). Unassigned and signals were excluded from fragmentation, and dynamic exclusion was set to 35 s for 2–6× charged features.

### 4.11. Protein and Phosphopeptide Identification and Quantification

The MS files were processed with MaxQuant software (version 1.6.14; Max Planck Institute of Biochemistry, Munich, Germany) [46] and searched with the Andromeda search engine [47] against the human UniProt database (sequences 74,830 entries). Enzyme specificity was set to that of trypsin, allowing for cleavage of the N-terminal to proline residues and up to 2 missed cleavage sites (for proteome) or up to 4 missed cleavage sites (for phosphopeptides). A minimum peptide length of 7 amino acids was required. Carbamidomethylation (C) was set as a fixed modification, whereas oxidation (M), deamidation (NQ), protein N-terminal acetylation and, if necessary, phosphorylation (STY) were considered as variable modifications. No labeling or double SILAC labeling was defined according to a maximum of 3 or 5 labeled amino acids. Mass tolerances were defined for the precursor and fragmented ions as follows: MS first search—20 ppm, MS main search—6 ppm and MS/MS deisotoping tolerance—0.7 ppm. The false discovery rates (FDRs) at the protein and peptide levels were set to 1%. SILAC-based quantification was based on unique and razor peptides only, and a minimum of two ratio counts was required. Peptide ratios were calculated and normalized for each arginine- and/or lysine-containing peptide as described [46]. In addition, the “match between the runs” feature was implemented with default settings to increase the number of quantified peptides.

Further data analysis was performed in Perseus (version 1.6.15; Max Planck Institute of Biochemistry, Munich, Germany). Matches to the reverse database proteins identified by one site only in modified peptides and common contaminants (KRT2 and KRT10) were removed from the MaxQuant output. Exclusively, phosphosites quantified in at least 2 (out of 3) replicates and with a localization probability higher than 0.75 were subjected to further analysis. Only proteins identified with at least two unique peptides and quantified in at least 2 (out of 3) biological replicates were considered for the subsequent analysis. The obtained phosphopeptides ratios were corrected for differential protein expression by dividing the matched protein ratios. Proteins and phosphosites changed by >1.5-fold in at least two biological replicates were considered regulated. In addition, to each identified and quantified phosphosite, information from the PhosphoSite Plus database [48] was assigned, including known phosphosites. 

### 4.12. Data Analysis

A global interaction network of regulated proteins and phosphoproteins was predicted in STRING v11.5 (available at: www.string-db.org (accessed on 1 April 2022)) [49]. Each protein–protein interaction (PPI) has a combined score (edge score), which represents the reliability of the interaction between proteins. Settings included (i) 40 seed proteins, (ii) human genome interactions, (iii) a highest confidence PPI interaction score of 0.9 and (iv) 500 first shell nodes, resulting in a PPI-enrichment *p*-value of <1 × 10^−16^. In addition, an enrichment analysis of the regulated proteins and phosphoproteins were also performed in STRING v11.5 for the Gene Ontology Cellular Compartments (GOCC) and Molecular Function (GOMF). Cluster analysis was based on k-means clustering and restricted to 8 clusters. Multiple hypothesis testing was controlled by using a Benjamini–Hochberg FDR. Visible clusters on PPI maps were assigned by color coding the nodes.

### 4.13. Metabolic Labeling and Immunoprecipitation of Proteins

For the metabolic labeling of cellular proteins, cells were grown in the presence of [^35^S]-methionine and [^32^P]-orthophosphate to test the protein phosphorylation, as described previously [9]. 

Immunoprecipitation was performed using protein A/G-presenting magnetic beads (Thermo Fisher Scientific, Darmstadt, Germany) as described before [9] using 1 to 2 mg of cell protein lysate in RIPA buffer with appropriate amounts of the following antibodies: anti-PURB (Thermo Fisher Scientific, Darmstadt, Germany): 5 µg; anti-ZBTB7A (Abcam, Cambridge, UK): 5 µg; CKbeta2 (Thermo Fisher Scientific, Darmstadt, Germany: 4.2 µg; FOXK1 (Cell Signaling Technology, Danvers, MA, USA): 3 µg; BTF3 (Abcam, Cambridge, UK): 8.5 µg; BCAR1 (OriGene Technologies GmbH, Herford, Germany): 4 µg and MAPKAPK3 (Cell Signaling Technology, Danvers, MA, USA): 6 µL (no protein concentration given by the supplier). Elution was carried out with 12 µL HCl·glycine (pH 2.5) solution. Eluates were mixed with 10 mL scintillation cocktail (UltimaGold; PerkinElmer, Waltham, MA, USA), and radioactivity was determined in a liquid scintillation counter (Tricarb 2900, PerkinElmer, Waltham, MA, USA), applying the transformed Spectral Index of the External Standard/Automatic Efficiency Control method. 

## Figures and Tables

**Figure 1 ijms-23-06414-f001:**
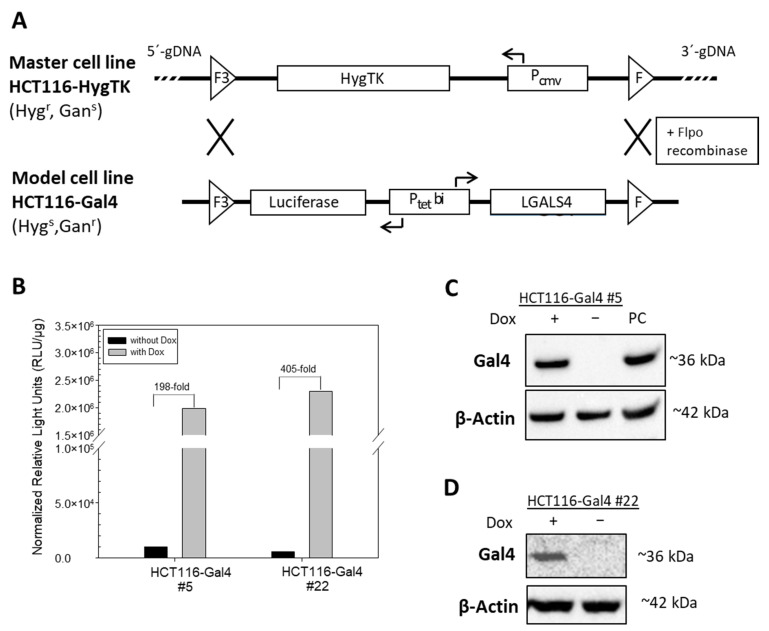
Generation and characterization of HCT116-Gal4 cells by recombinase-mediated cassette exchange (RMCE). (**A**) The master cell line HCT116-HygTK was used to generate the model cell line, HCT116-Gal4 by RMCE. Wildtype (F) and mutant (F3) 2 Flpo-recombinase recognition sites, as well as the direction of the transcription (arrows) of cassette genes (*LGALS4*, HygTK and luciferase), by constitutive (P_CMV_) and dox-inducible, bidirectional (P_tet_bi) promoters are indicated. Master and model cell lines differ in sensitivity (s) or resistance (r) towards the antibiotics hygromycin (Hyg) and ganciclovir (Ganc). (**B**) Induction of luciferase activity by treatment with doxycycline. HCT116-Gal4 #5 and #22 cells were cultured in the presence or absence of 0.5 μg/mL Dox for 24 h. (**C**,**D**) Induction of Gal4 on the protein level analyzed by Western blot. The cell lysate from the Gal4-positive cell line Colo 205 served as the positive control (PC), and the beta-Actin signal was used as the loading control.

**Figure 2 ijms-23-06414-f002:**
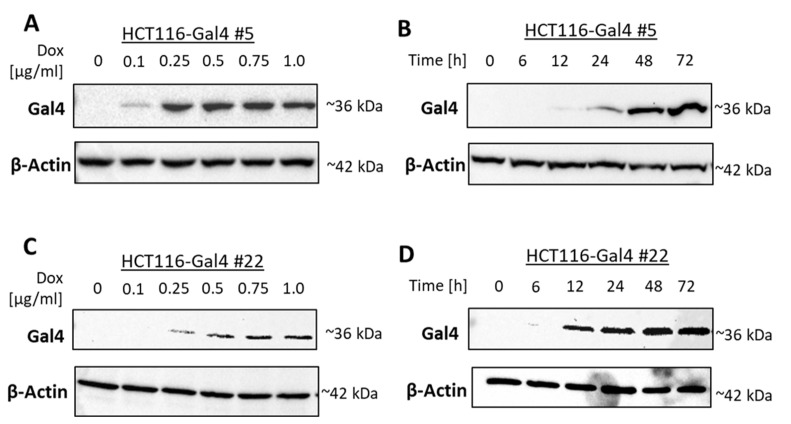
Dose- and time-dependent induction of Gal4 protein expression. HCT116-Gal4 #5 and #22 cells were cultured either in the presence or absence of the indicated concentrations of Dox for 24 h (**A**,**C**; dose–response). In time–course experiments performed for up to 72 h, different Dox concentrations were chosen for each clone to ensure a strong expression: 0.5 μg/mL (clone #5) or 0.75 μg/mL Dox (clone #22) (**B**,**D**).

**Figure 3 ijms-23-06414-f003:**
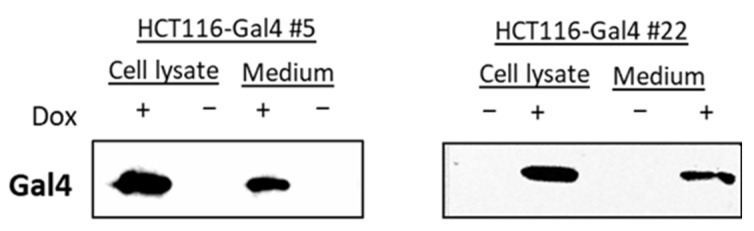
Gal4 secretion analysis. Concentrated medium from doxycycline-treated HCT116 Gal4 cells and untreated controls were analyzed together with the respective cell extracts (50 µg) by Western blotting.

**Figure 4 ijms-23-06414-f004:**
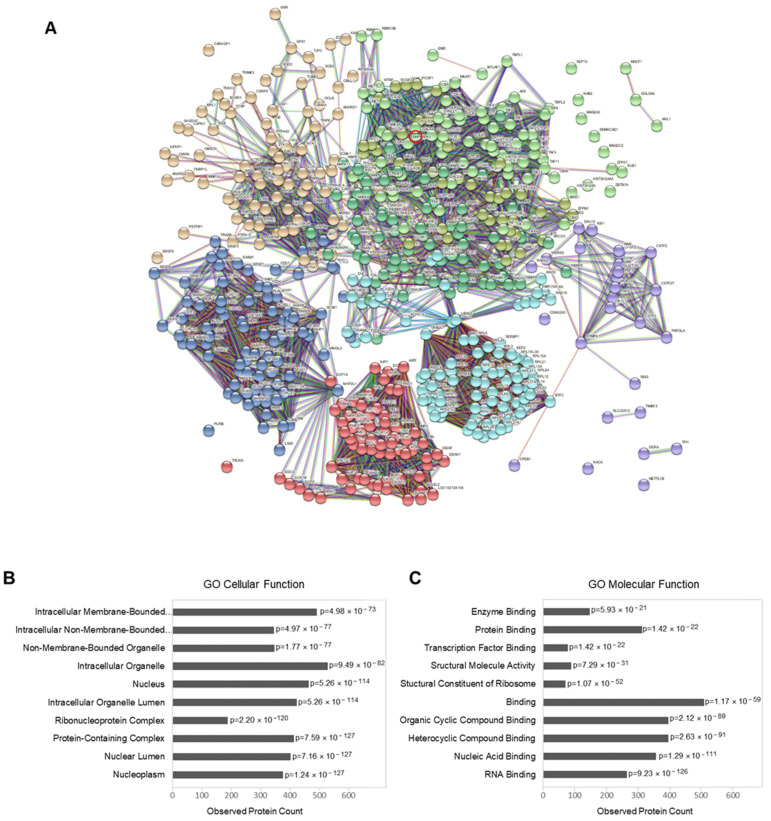
Phosphoproteomic data analysis: (**A**) interaction network of regulated proteins and phosphoproteins (with at least one regulated phosphorylation site) upon Gal4 expression (generated by STRING v11.5; EMBL, Heidelberg, Germany). The connecting lines between protein nodes represent protein–protein interactions based on an interaction score of the highest confidence (0.9). Coloring of the lines indicates the type of interaction evidence, and coloring of the proteins is based on a cluster analysis restricted to a specified number of k-means clusters (*n* = 8). Gal4 (*LGALS4*) is marked by a red circle. (**B**,**C**) Enrichment analysis of the regulated proteins (performed in STRING v11.5). Graph showing the most enriched Gene Ontology (GO) cellular components (**B**) and molecular functions (**C**) among the regulated proteins and phosphoproteins, with the observed protein count in each category and calculated *p*-value corrected for multiple testing according to Benjamini and Hochberg.

**Figure 5 ijms-23-06414-f005:**
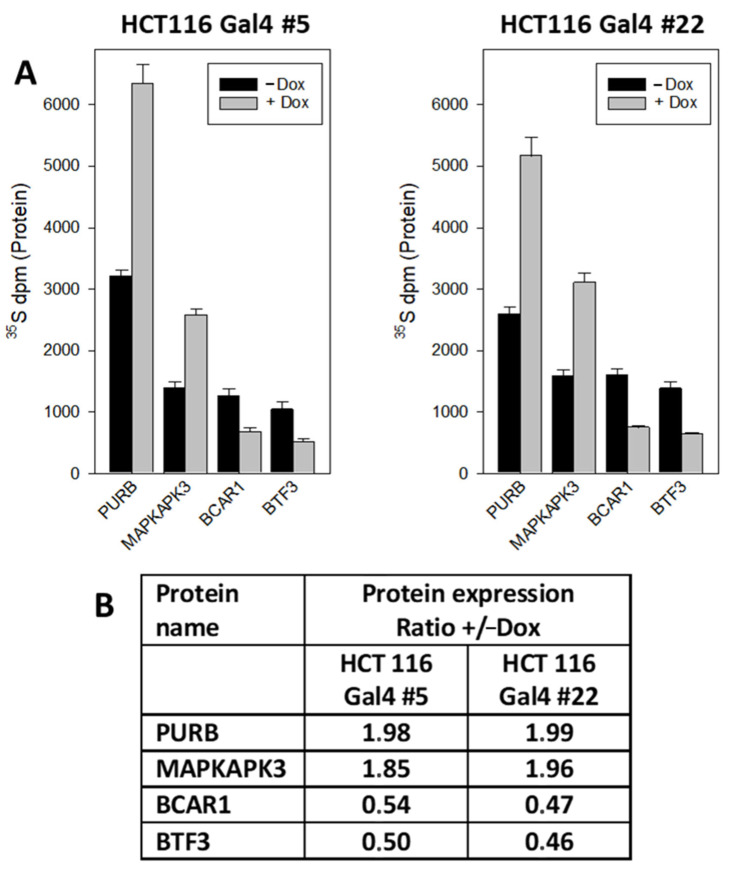
Quantification of Gal4-regulated proteins by radiometric metabolic labeling and immunoprecipitation: (**A**) cellular proteins in HCT116 Gal4 clones #5 and #22 were metabolically labeled with ^35^S-methionine in the culture medium in the presence or absence of dox for 72 h. Then PURB, MAPKAPK3, BCAR1 and BTF3 were immunoprecipitated and the radioactivity counted in the precipitates. The results are the mean of three independent observations (+/−S.D.). (**B**) Relative change in the protein expression: +Dox vs. −Dox.

**Figure 6 ijms-23-06414-f006:**
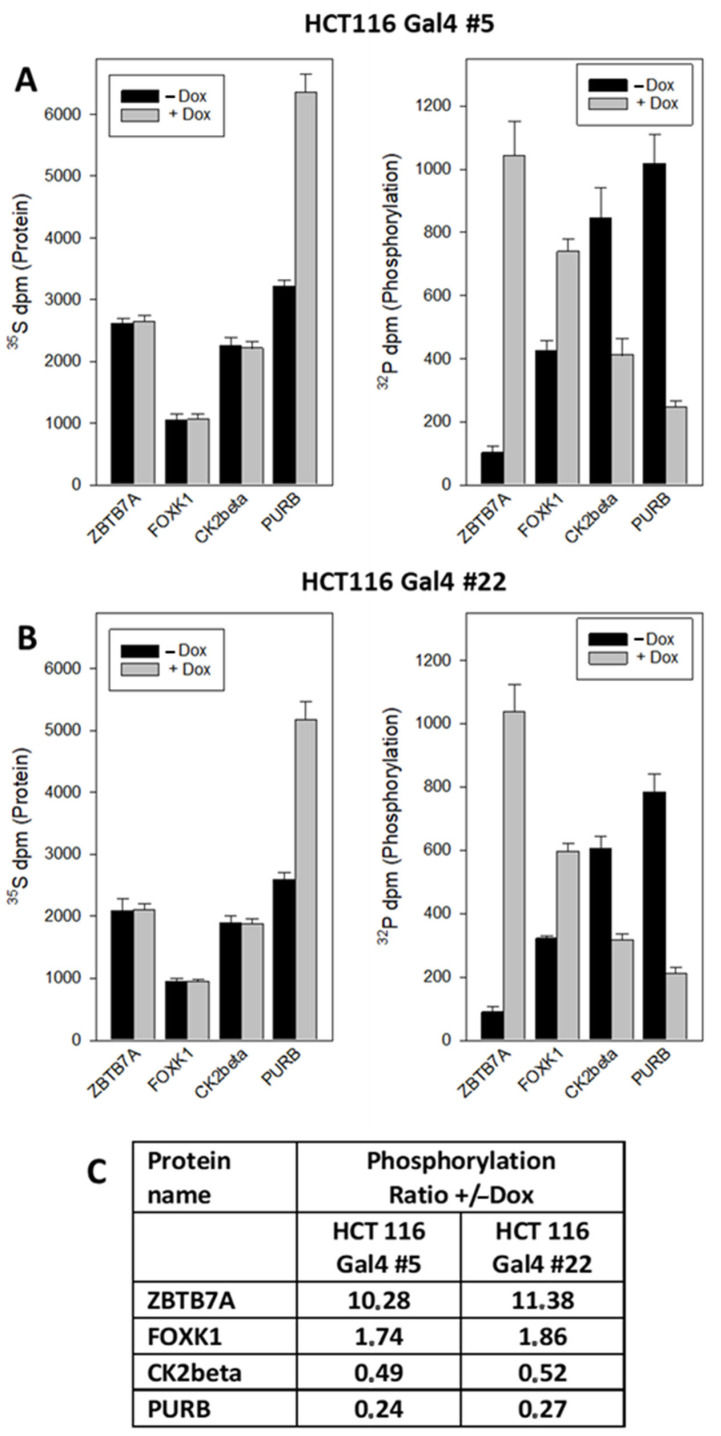
Quantification of Gal4-induced phosphorylation changes by dual radiometric labeling and immunoprecipitation. (**A**,**B**) Cells were treated as described in Figure 5. Additionally, ^32^P-orthophosphate was added to the culture medium for the last hour of treatment. Then, ZBTB7A, FOXK1, CK2beta and PURB were immunoprecipitated, and the radioactivity of both nuclides counted in the precipitates. The results are the mean of three independent observations (+/−S.D.). (**C**) Relative change in the phosphate incorporation: +Dox vs. −Dox.

**Table 1 ijms-23-06414-t001:** Altered protein expression upon Gal4 re-expression.

Proteins	Genes	Ratio ^1^ +/− Dox Rep1	Ratio ^1^ +/− Dox Rep2	Ratio ^1^ +/− Dox Rep3	Mean Ratio
Galectin-4	*LGALS4*	12.16	21.05	44.36	25.86
Glutathione peroxidase	*GPX1*	1.99	2.20	1.92	2.04
Calcium-binding mitochondrial carrier protein Aralar1	*SLC25A12*	3.20	1.11	1.65	1.98
Transcriptional activator protein Pur-beta	*PURB*	1.83	2.10	1.98	1.97
MAP kinase-activated protein kinase 3	*MAPKAPK3*	1.78	1.93	NaN	1.85
15 kDa selenoprotein	*SEP15*	1.65	2.02	1.63	1.77
Polyhomeotic-like protein 2	*PHC2*	1.48	1.72	1.57	1.59
Ribosomal protein S6 kinase	*RPS6KA4*	NaN	1.61	1.57	1.59
Deoxyribose-phosphate aldolase	*DERA*	NaN	1.57	1.56	1.56
Titin	*TTN*	2.59	1.85	0.23	1.56
Methyltransferase-like protein 7B	*METTL7B*	0.36	0.33	1.49	0.73
Breast cancer anti-estrogen resistance protein 1	*BCAR1*	0.69	0.59	0.57	0.62
Transcription initiation factor IIB	*GTF2B*	0.59	NaN	0.63	0.61
Golgin subfamily A member 4	*GOLGA4*	NaN	0.55	0.64	0.60
WD repeat-containing protein 46	*WDR46*	0.43	0.97	0.35	0.58
Ubiquitin carboxyl-terminal hydrolase 13	*USP13*	0.62	0.54	NaN	0.58
Gamma-taxilin	*TXLNG*	0.87	0.63	0.21	0.57
Zinc finger CCCH domain-containing protein 13	*ZC3H13*	0.51	NaN	0.55	0.53
Transcription factor BTF3	*BTF3*	NaN	0.44	0.57	0.50
Tyrosine-protein phosphatase non-receptor type 12	*PTPN12*	0.60	0.40	NaN	0.50

^1^ NaN: Not a Number.

**Table 2 ijms-23-06414-t002:** Top regulated phosphosites upon Gal4 re-expression.

Protein Names	Gene Names	Amino Acid Position	Ratio +/− Dox Rep1	Ratio +/− Dox Rep2	Ratio +/− Dox Rep3	Mean Ratio
Zinc finger and BTB domain-containing protein 7A	*ZBTB7A*	S549	13.52	8.80	5.27	9.20
Tumor suppressor p53-binding protein 1	*TP53BP1*	S500	2.46	2.98	4.56	3.33
Microtubule-associated protein tau	*MAPT*	S133	3.28	2.69	1.33	2.43
Enhancer of mRNA-decapping protein 4	*EDC4*	S723	1.55	2.18	1.28	1.67
Tight junction protein ZO-3	*TJP3*	S327	2.01	1.23	1.65	1.63
Serine/arginine repetitive matrix protein 1	*SRRM1*	S883	0.93	2.33	1.59	1.62
Serine/arginine repetitive matrix protein 1	*SRRM1*	T881	0.93	2.33	1.59	1.62
Serine/arginine repetitive matrix protein 2	*SRRM2*	S2121; S2123	0.70	1.94	1.94	1.53
Calcium-regulated heat stable protein 1	*CARHSP1*	S30; S32; S41	1.48	1.54	1.51	1.51
Serine/arginine repetitive matrix protein 2	*SRRM2*	S876	0.97	1.91	1.60	1.49
Serine/arginine repetitive matrix protein 2	*SRRM2*	S875; S876	0.80	1.85	1.78	1.47
Serine/arginine repetitive matrix protein 2	*SRRM2*	S2692; S2694	0.82	1.72	1.73	1.42
Forkhead box protein K1	*FOXK1*	S441; S445	1.02	1.61	1.58	1.40
Forkhead box protein K1	*FOXK1*	T436	1.02	1.61	1.58	1.40
Serine/arginine-rich splicing factor 9	*SRSF9*	S211; S216	0.62	1.63	1.79	1.35
Serine/arginine repetitive matrix protein 2	*SRRM2*	S1727	0.78	1.61	1.61	1.33
Serine/arginine repetitive matrix protein 2	*SRRM2*	S954	0.87	1.57	1.51	1.32
Serine/arginine repetitive matrix protein 2	*SRRM2*	S952	0.86	1.52	1.51	1.30
Serine/arginine repetitive matrix protein 1	*SRRM1*	S384; S386; S388	0.76	1.51	1.58	1.28
Nucleolin	*NCL*	S41; S42	1.23	0.55	0.62	0.80
General transcription factor IIF subunit 1	*GTF2F1*	S221	1.19	0.54	0.63	0.78
Bcl-2-associated transcription factor 1	*BCLAF1*	S225; S228	1.23	0.47	0.62	0.77
General transcription factor IIF subunit 1	*GTF2F1*	S224	1.08	0.58	0.58	0.75
Chromodomain-helicase-DNA-binding protein 4	*CHD4*	S308; S309; S310	1.03	0.60	0.52	0.72
Plasminogen activator inhibitor 1 RNA-binding protein	*SERBP1*	S330	0.79	0.65	0.62	0.69
Transcription intermediary factor 1-beta	*TRIM28*	S473	0.71	0.59	0.60	0.64
Casein kinase II subunit beta	*CSNK2B*	S228	0.63	0.66	0.61	0.63
Formin-binding protein 1-like	*FNBP1L*	S488	0.28	0.64	0.95	0.62
Segment polarity protein dishevelled homolog DVL-2	*DVL2*	S205	0.63	0.56	0.59	0.60
Formin-binding protein 1-like	*FNBP1L*	S488; S489	0.22	0.56	0.99	0.59
Transcriptional activator protein Pur-beta	*PURB*	S6; S8	0.54	0.65	0.52	0.57
Transcriptional activator protein Pur-beta	*PURB*	S6	0.57	0.43	0.53	0.51
Symplekin	*SYMPK*	S494	0.60	0.39	0.49	0.49

## Data Availability

The mass spectrometry proteomics data have been deposited to the ProteomeXchange Consortium via the PRIDE [50] partner repository with the dataset identifier PXD032902.

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
