# Peer review of "Combining Recombinase-Mediated Cassette Exchange Strategy with Quantitative Proteomic and Phosphoproteomic Analyses to Inspect Intracellular Functions of the Tumor Suppressor Galectin-4 in Colorectal Cancer Cells"

_ijms, 2022, doi:10.3390/ijms23126414_

Round 1
Reviewer 1 Report
Manuscript No. ijms-1691960
„Combining Recombinase-Mediated Cassette Exchange Strategy with Quantitative Proteomic and Phosphoproteomic Analyses to Inspect Intracellular Functions of the Tumor Suppressor Galectin-4 in Colorectal Cancer Cells” for International Journal of Molecular Sciences
Comments:
- Introduction. I am asking the Authors to formulate a clear and unambiguous purpose of the work at the end of this section. Was the goal to develop a line or to perform the analyzes indicated by the Authors, or to compare the created clones?
- Materials and methods 4.1. Please provide penicillin concentration as U/mL.
- Materials and methods 4.1. Please provide catalog numbers of the cultures used. At the moment, the presented criterion of the general name can be met by several lines.
Author Response
Point 1
The reviewer asked the authors to formulate a clear and unambiguous purpose of the work at the end of this section.
We have stated the purpose of this study more clearly by rephrasing the last sentence of the Introduction (page 2, lines 78-82).
Point 2
Penicillin concentrations should be provided as U/ml in Materials and methods.
The antibiotic penicillin has been used at 100 U/ml in all cell culture media which is now correctly indicated in Materials and methods section 4.1 (page 12, line 325).
Point 3
Catalog numbers of cell lines should be provided.
In addition to the general names we now have added the ECACC catalog numbers as unique identifiers of the cell lines HCT116, Colo205 and HeLa (page 12, lines 327-327).
Reviewer 2 Report
This referee finds the experimental design implemented by the authors well organized and the experiments well designed and conducted with appropriate technologies and methods. In particular, I appreciated the method used to identify the effects of Gal4 on other proteins; I mean the use of doxycycline and detection by isotope labeling and mass.
Therefore, I agree with the proteins that have been selected. Many of them have phosphorylation sites because they are IDPs or mixed IDPs (organized part + disordered segments (IDR)). You can check their 3D structure on AlphaFold. As is known, these segments show a high probability of receiving phosphorylation sites and give the protein great structural flexibility. Structural flexibility + post-translational changes also mean a high probability that their many changes open up enormous possibilities for functional interactions.
Each modified form is an independent molecular structure (proteoform) very different from the native form (which, in fact, does not exist). Each proteoform has its own specific functional characterization that makes these proteins chameleonic. However, often we do not know where, when, and how each proteoform reacts functionally. All these multiple functions should not be added to a single molecular entity (the native protein) when it acts through many chemically and structurally different forms. In this way, we alter the interacting capacity of the node (node degree) because the proteoform functions that are present in different functional/experimental contexts are also collapsed on it. This happens on STRING that, in the absence of information defining where, when, and how each proteoform operates, collapses all known functions on the native node. This alters the properties of the node because it gives it an excess of space-time interactions that the proteoform is not exercising (increases the value of the node degree).
The authors should take this reflection into account.
They conveniently used STRING to discover the functional role of their proteins and in fig 4 they show the functional network.
The authors used a confidence score of 0.4., and the 41 proteins as interacting seeds. But the graph shows only 25 nodes, with 35 edges, an avg-local-clustering coefficient of 0.332, and a PPI enrichment value of 3.8e-5. They also show an enrichment of GO Cellular Components and Molecular functions. Unfortunately, the statistical parameters of the network are poor. A functional network rests its statistically significant structure on functional information extracted from the archives that contain it. It is easy to check that only a few interactions (9/35 interactions) of the network are derived from the STRING channel which extracts the experimentally proven interactions. Most of the interactions come from the channel managed by text-mining or other indirect less reliable methods. The Authors do not explain the fact that 16 proteins are not functionally connected. This means that they are not functionally involved in the pathological model they are studying.
Any biomolecule that is hypothesized to be involved in a pathology does not carry out its function alone but does it within the functional module of the metabolic network that characterizes the pathology itself and contains all the other biomolecules functionally involved with it. When we experimentally identify several biomolecules as present in a pathological context, nothing tells us they are all functionally involved in the pathology. They are present in the pathological tissue but might not be involved in any characteristic function of the disease.
Another fact to consider is that we only see interactions that are already present in the STRING archives. Therefore, a study that for the first time discovers new functions for certain proteins may not have an inadequate networking analysis, simply because the archive ignores their presence. The low score (0.4) also increases uncertainty.
Drawing the conclusions, the analysis presented is statistically weak and therefore with poorly reliable functional information (low number of nodes and of edges, as well the average node degree). In summary, we should say that all the results are statistically unreliable, which would suggest an inadequate experimental design with many false results. In fact, the test conditions show involvement of these proteins as colonic cancer cells or colon with an FDR value of 0.0027 and 0.0495, respectively.
Fortunately, this is not the case because notwithstanding the networking analysis shown is naïve (I apologize for the term but it is adequate) can be improved.
I suggest the authors use the more reliable confidence score of 0.9. To use the 41 proteins as seeds to extract all interactions from the entire human genome up to show a PPI enrichment p-value of 1e-16. To do this, it is necessary to enrich the number of interactors (under Settings) up to 500 first shell nodes (under Custom value), that is direct interactions of the first order. As you can see the system accommodates all enrichment.
Under this setting, you will have 541 nodes, 9808 edges, an average node degree of 36.3 but above all a PPI enrichment p-value <1.0e-16 (with Cytoscape the analysis may also have a greater number of nodes that is useful to calculate the power law, where its graphical representation is the proof that we are dealing with a real biological net). You also find that you now have 1156 GO terms for Biological Processes, 140 for Molecular functions, 122 KEGG paths, and so on (for 2991 enriched terms).
But, now we also know that all your proteins work in the nucleus (1.54e-136), are all phosphoproteins (2.09e-63), are involved in complexes (2.08e-128) among which the ribonucleoprotein complex (4.4e-128), in mRNA processing (9.27e-35), in mRNA metabolism (5.0e-177), in colon cancer (4.2e-11) or colon (1.76e-08) just to mention some functional and pathological aspects. From direct interactions, you will see that LGALS4 might be directly involved in the metabolism of fatty acids and in the control of glucose homeostasis, ...... .. ...... and many other considerations.
A clustering analysis may implement also the study. In this way, you will have a robust and meaningful System of Analysis that will allow you to say much more than what you have said, endorsing your experimental design and improving and expanding your findings. And above all, you will be more precise in describing the proteins really involved and their functional properties without having to make an endless list of all of them with the alleged activities taken from the literature.
An excellent proteomics analysis followed by a good interactomics analysis opens many ajar windows. I advise you to improve the interactomics analysis.
Author Response
The reviewer expressed serious concern about the poor statistical parameters and settings of our STRING-based network analysis. He suggested a cluster analysis and asked for improved interactomics analysis.
We completely agree with the reviewer. His critical reflections and detailed setting suggestions were extremely helpful to us and are gratefully appreciated. Using these settings we now provide a new graph of the network (Fig4A) that also displays specific clusters. The highest enriches categories of a new Gene Ontology analysis are indicated in revised Fig4B and C and are reflected by high statistical values. Even with this improved interactomics analysis, the main results and conlusions of the original manuscript still remain valid. We have adapted the legend to Fig 4 as well as the Results (page 7) and Materials and methods section 4.12 (page 16), accordingly.
Round 2
Reviewer 2 Report
I thank the authors for appreciating my criticism, aimed at improving the meaning of your work.
Good work.